# Record ages of non-Markovian scale-invariant random walks

Léo Régnier[1], Maxim Dolgushev [1] & Olivier Bénichou [1]✉

How long is needed for an observable to exceed its previous highest value and establish a new record? This time, known as the age of a record plays a crucial role in quantifying record statistics. Until now, general methods for determining record age statistics have been limited to observations of either independent random variables or successive positions of a Markovian (memoryless) random walk. Here we develop a theoretical framework to determine record age statistics in the presence of memory effects for continuous non-smooth processes that are asymptotically scale-invariant. Our theoretical predictions are confirmed by numerical simulations and experimental realisations of diverse representative non-Markovian random walk models and real time series with memory effects, in fields as diverse as genomics, climatology, hydrology, geology and computer science. Our results reveal the crucial role of the number of records already achieved in time series and change our view on analysing record statistics.

The statistics of records in a discrete time series $(X_t)_{t=0,1,\dots}$ is one of the main topics of interest in the study of extreme events[1], with applications in an increasing number of fields. A record event occurs at time $t$ if all prior observations $(X_{t'})_{t'=0,\dots,t-1}$ are smaller than the last value $X_t$. In this context, the inter record times $\tau_n$, also called record ages[2–9], between the $n^{th}$ and $(n+1)^{st}$ record, are pivotal, as they characterise the time of occurrence of the next record breaking event such as heatwaves[10], earthquakes[11,12] or record temperatures[13].

The theory of records has been studied since the mid-20th century[14,15], and is well understood when the random variables $(X_t)_{t=0,1,\dots}$ are independent and identically distributed (i.i.d.)[16–18]. An important step in the study of records was recently made when observations are the successive positions of a Markovian RW[4,19–22], $X_{t+1}=X_t+\eta_{t+1}$, where the steps $(\eta_t)_{t=0,1,\dots}$ are still i.i.d. and symmetric. In this situation, record ages are strictly given by the time $T$ needed to reach a given value for the first time, regardless of the past. This time follows an algebraic tail distribution $\mathbb{P}(T \geq \tau) \propto \tau^{-\theta}$, where $\theta$ is the persistence exponent[23], provided by the celebrated Sparre-Andersen theorem[24], yielding $\theta=1/2$. We emphasise that, despite the fact that this RW model accounts for correlations between the observations $(X_t)_{t=0,1,\dots}$, the steps $(\eta_t)_{t=0,1,\dots}$ themselves are independent. As a

result, this model cannot account for memory effects in the increments.

However, as a general rule, real time series are not only correlated but also exhibit such memory effects. When the evolution of an observable is influenced by interactions with hidden degrees of freedom, such as the previous steps of the RW or its interaction with the environment, it cannot be modeled as a Markov process.

This is typically the case for displacement data from various tracers (microspheres, polymers, cells, vacuoles...) in simple[25] and viscoelastic fluids[26–28], soil[29,30] and air temperatures[31], river flows[32,33], nucleotide sequence locations[34,35] and Ethernet traffic[36–38]. So far, as highlighted in the recent review Ref. 4, almost nothing is known about the record age statistics of non-Markovian processes. The only exceptions concern processes amenable to a Markovian process by adding an extra degree of freedom[3,8,39], and a numerical observation in the specific case of the fractional Brownian motion[9]. Here, we provide a general scaling theory which determines the time dependence of the record age statistics of non-Markovian RWs. We show that memory effects significantly alter these statistics. They are no longer solely governed by the persistence exponent $\theta$, but also by another explicitly calculated exponent, which is the hallmark of non-Markovian dynamics.

[1]Laboratoire de Physique Théorique de la Matière Condensée, CNRS/Sorbonne Université, 4 Place Jussieu, 75005 Paris, France.
✉e-mail: benichou@lptmc.jussieu.fr

## Results

### Main results

We consider a general non-Markovian symmetric RW, whose successive positions form a time series $(X_t)_{t=0,1,\ldots}$. These positions satisfy $X_{t+1} = X_t + \eta_{t+1}$, where now the statistics of the steps $(\eta_t)_{t=0,1,\ldots}$ may exhibit (*I*) long-range correlations, (*II*) interactions with the environment (e.g. footprints left along the trajectory), or (*III*) explicit space-time dependence (see Fig. 1). Essentially all statistical mechanisms that lead to non-Markovian evolution are encompassed by these features of $X_t$[40]. In turn, they allow to account for a variety of real time series displaying memory effects[41,42]. At large time, $X_t$ is assumed to converge to a scale-invariant process that is continuous (i.e. excluding broadly distributed steps $\eta_t$) and non-smooth[23] (meaning that, as for the standard Brownian motion, the trajectory is irregular, having at each point an infinite derivative). Under these conditions, the process is characterised by a walk dimension[40] $d_w > 1$, such that $X_t \propto t^{1/d_w}$, and the random variable $X_t/t^{1/d_w}$ is asymptotically independent of $t$. To account for potential aging in the increments, $X_t$ is more generally assumed to have scale-invariant increments, meaning that, for $1 \ll t \ll T$, $X_{t+T} - X_T \propto t^{1/d_w^0} T^{\alpha/2}$. This defines the aging exponent $\alpha$[43,44] ($\alpha > 0$ corresponding qualitatively to accelerating processes and $\alpha < 0$ to slowing down processes) and an effective walk dimension at short times $d_w^0 \equiv (d_w^{-1} - \alpha/2)^{-1}$. We stress that the class of processes that we consider here covers a very broad range of examples of non-Markovian RWs, as detailed below, despite not covering the particular cases of Lévy flights[19] (which are discontinuous) or of the Random Acceleration Process[3] (smooth), which would require a different approach.

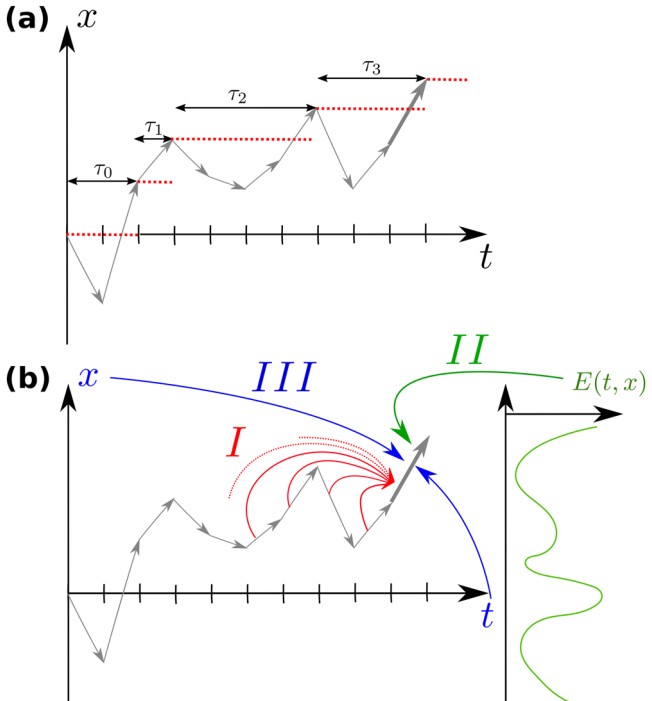

**Fig. 1 | Record ages for non-Markovian random walks (RWs). a** Sketch of a space time trajectory of the RW represented by successive discrete steps $\eta_t$ (grey arrows). The records in the trajectory are identified by red dotted lines. The record age $\tau_n$ of the RW is defined as the time between the $n^{th}$ and $(n+1)^{st}$ records. **b** Different statistical mechanisms giving rise to a non-Markovian evolution: The statistics of the RW steps $\eta_t$ may depend on (*I*) the previous steps of the walk (red arrow), (*II*) the environment with which the RW interacts (green arrow, schematically represented by the function $E(t,x)$), or (*III*) the current time or position (blue arrows). In this article, we show that these memory effects strongly modify the record age statistics, which are no longer simply given by the usual persistence exponent $\theta$, but also by a distinct exponent that we determine explicitly.

We report that the tail distribution $S(n,\tau) \equiv \mathbb{P}(\tau_n \geq \tau)$ of the record age $\tau_n$ asymptotically obeys a scaling behaviour $S(n,\tau) = n^{-1} \psi(\tau/n^{d_w})$, displaying two universal distinct algebraic regimes :

$$S(n,\tau) \propto \begin{cases} \frac{1}{n}\left(\frac{n^{d_w}}{\tau}\right)^{\frac{1}{d_w^0}} & \text{for } n^{d_w - d_w^0} \ll \tau \ll n^{d_w}, \\ \frac{1}{n}\left(\frac{n^{d_w}}{\tau}\right)^{\theta} & \text{for } 1 \ll n^{d_w} \ll \tau \end{cases} \quad (1)$$

where $\psi$ is a process dependent scaling function and the persistence exponent $\theta$ has been defined above. Equation (1) explicitly determines the $n$ and $\tau$ dependence of the record age statistics of non-Markovian RWs. Fundamental consequences of our results include: (i) In regime 1, defined by $n^{d_w - d_w^0} \ll \tau \ll n^{d_w}$, the record time's decay is governed by an exponent different from $\theta$. While it is not unexpected that the memory of the past affects record age statistics for a non-Markovian process (in particular, it is known that it can change the persistence exponent[45,46]), it is striking that the corresponding exponent is fully explicit and depends only on the effective walk dimension $d_w^0$ of the increments. Note that regime 1 can span several orders of magnitude as soon as sufficiently many records have been broken, and thus dominate the observations. (ii) In regime 2, defined by $\tau \gg n^{d_w}$, the decay in the record time can be very different from that of regime 1. This is particularly striking for processes with stationary increments for which the exponent involved in regime 2, $\theta = 1 - 1/d_w$[44], is markedly different from the exponent $1/d_w^0 = 1/d_w$ of regime 1 (with the exception of Markovian RWs for which the two exponents are both 1/2 and a single regime is recovered; note that this single regime of exponent 1/2 is also obtained in the case of Lévy flights, which are not covered by our approach). (iii) The record age distribution ages, in the sense that it depends on the number $n$ of records already achieved. Consequently, the observations of early record ages are not representative of later records and call for a careful analysis of real data (note that the record distribution also ages in time series with i.i.d. observations $X_t$, which are thus not of the form $X_{t+1} = X_t + \eta_{t+1}$ considered here, but the dependence of this distribution on the number of records and the corresponding statistical mechanisms are very different[4]). Finally, note that despite the existence of two regimes for record ages, because of the explicit dependence of the prefactors of $S(n,\tau)$ on $n$, the number of records at time $t$ displays a single time regime $n \propto t^{1/d_w}$ (see Supplementary Information, SI).

### Derivation of the results

The following outlines the derivation of these results (see SI Sec. S1 for details):

The first step consists in noting that, due to the scale-invariance of the process $X_t$, the time $T_n$ to reach the $n^{th}$ record, $T_n \equiv \sum_{k=0}^{n-1} \tau_k$, satisfies $T_n \propto n^{d_w}$ and its increments obey $T_{m+n} - T_m \propto m^{d_w - d_w^0} n^{d_w^0}$ (see SI Sec. S1.B). In other words, $\mathbb{P}(T_{m+n} - T_m \leq T)$ is a function of a single variable $T/(m^{d_w - d_w^0} n^{d_w^0})$. Then, $T_{m+n} - T_m = \sum_{k=m}^{n+m-1} \tau_k$ is dominated by the largest record age[40,47] under the self-consistent assumption that $S(n,\tau) \propto n^{-1+\epsilon_1} \tau^{-y_1}$ for $\tau \ll n^{d_w}$ (regime 1) and $S(n,\tau) \propto n^{-1+\epsilon_2} \tau^{-y_2}$ for $\tau \gg n^{d_w}$ (regime 2) with $y_i$ between 0 and 1. This results in the equation

$$\mathbb{P}(T_{m+n} - T_m \leq T) \simeq \mathbb{P}(\max(\tau_m, \ldots, \tau_{m+n-1}) \leq T). \quad (2)$$

Adapting the argument of Ref. 48, we show for continuous scale-invariant non-smooth processes analytically (see Sec. S1.D of SI) and verify numerically (see Sec. S2.C of SI) that, in Eq. (2), the record ages $\tau_k$

are asymptotically ($n \gg 1$) effectively independent, which leads to

$$\mathbb{P}(T_{m+n} - T_m \le T) \simeq \prod_{k=m}^{n+m-1} (1 - S(k,T)) . \quad (3)$$

First, for time scales $T$ much smaller than the typical time $T_m \propto m^{d_w}$ required to break $m$ records and for $n \ll m$ (regime 1), Eq. (3) becomes

$$\mathbb{P}(T_{m+n} - T_m \le T) \underset{T, n^{d_w} \ll m^{d_w}}{\propto} \exp\left[-\frac{\mathrm{const}.n}{m^{1-\epsilon_1} T^{y_1}}\right]. \quad (4)$$

Using $T_{m+n} - T_m \propto m^{d_w - d_w^0} n^{d_w^0}$ gives the exponents of regime 1 as $y_1 = 1/d_w^0$ and $\epsilon_1 = d_w/d_w^0$.

Second, for $\tau \gg n^{d_w}$ (regime 2), the memory of the $n$ broken records no longer affects the algebraic time decay of $S(n,\tau)$, which is thus given by the persistence exponent $\theta = y_2$. Taking $m = 0$ in Eq. (3), we get

$$\mathbb{P}(T_n \le T) \propto \exp\left[-\mathrm{const}.n^{\epsilon_2}/T^\theta\right]. \quad (5)$$

Using $T_n \propto n^{d_w}$ leads to the exponent $\epsilon_2 = d_w\theta$.

## Comparison with numerical simulations of non-Markovian models

We confirm the validity of our analytical results in Fig. 2 by comparing them to numerical simulations of a broad range of representative RW examples, which illustrate the classes (*I*), (*II*), and (*III*) of non-

Markovianity discussed above. Specifically, we consider (see SI for precise definitions and Supplementary Table 1 for a summary of their characteristics): (*I*) (a) the fractional Brownian motion (fBm), a non-Markovian Gaussian process, with stationary increments given by $\langle(X_t - X_0)^2\rangle = t^{2H}$, where $H$ is the Hurst exponent; this paradigmatic model has been used repeatedly to account for anomalous diffusion induced by long-range correlations in viscoelastic fluids[26] as well as temporal series displaying memory effects[41,42]; (b) its extension to quenched initial conditions (qfBm), for which the statistics of increments is not stationary anymore, and which describes for instance the height fluctuations under Gaussian noise of an initially flat interface[44,45]; (c) the elephant RW (eRW)[49], for which the current step is drawn uniformly from all of the previous steps performed by the RW, and then reversed with probability $\beta$; (*II*) (d) The Self-Attractive Walk (SATW), (e) Sub-Exponential Self-Repelling Walk (SESRW) and (f) True Self-Avoiding Walk (TSAW) are prototypical examples of self-interacting RWs[50–53], for which the RW deposits a signal at each lattice site it visits and then has a transition probability depending on the number of visits to its neighbouring sites (see SI for precise rules), so that memory emerges from the interaction of the walker with the territory already visited; these RWs have been shown to be relevant in the case of living cells, where it was demonstrated experimentally that various cell types can chemically modify the extracellular matrix, which in turn deeply impact their motility[54]; (*III*) Two models involving an explicit spatial or temporal dependence of the steps: (g) the sub-diffusive (resp. (h) the superdiffusive) Average Lévy Lorentz model (subALL and supALL, respectively)[55–57] for which the transmission (resp. reflection) probability at every site decays algebraically with the

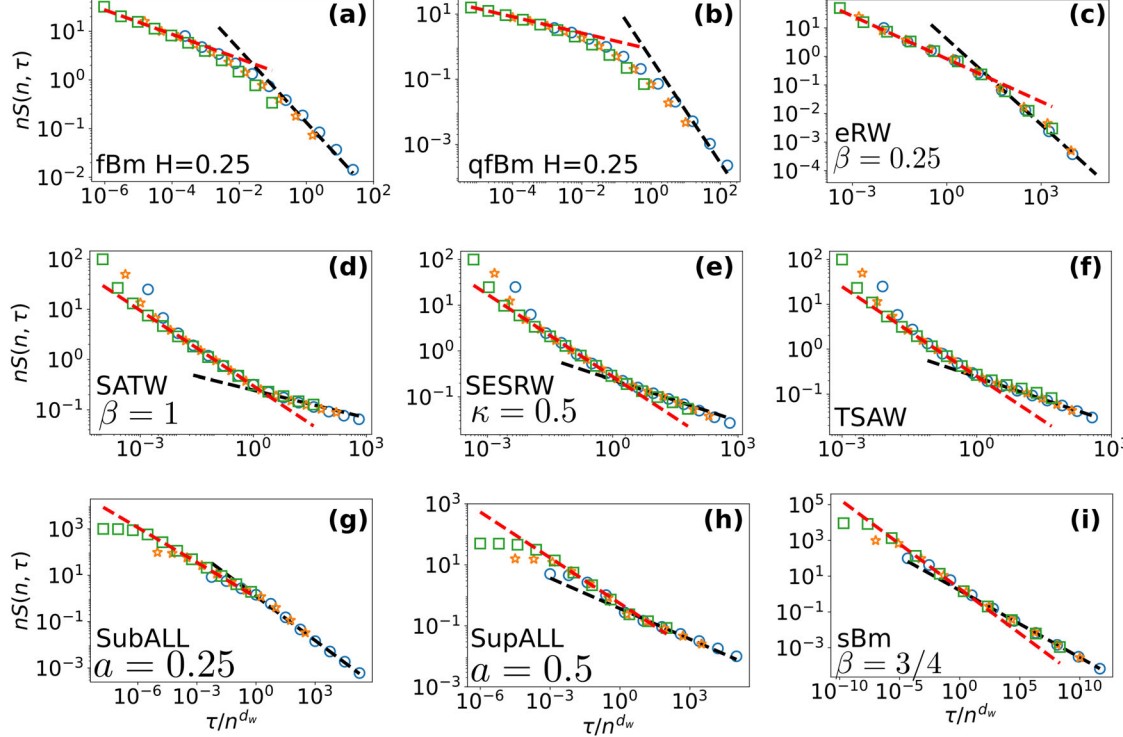

**Fig. 2 | Universal record age distributions for non-Markovian RWs: theoretical predictions (lines) vs numerical simulations (symbols).** Simulated rescaled tail distribution of record ages $\tau_n$ for different values of record number $n$ displayed for various representative RW models: (**a**) fractional Brownian motion (fBm) of Hurst exponent $H = 0.25 = 1/d_w = 1 - \theta$ for $n = 8, 16$ and $32$ (**b**) quenched fBm (qfBm) of Hurst exponent $H = 0.25 = 1/d_w$ and $\theta \approx 1.55$ for $n = 5, 10$ and $20$ (**c**) elephant RW (eRW) with $\beta = 0.25$ such that $d_w = 2$ and $\theta = 1$, for $n = 10, 25$ and $50$ (**d**) Self-Attractive Walk (SATW) with $\beta = 1$, such that $d_w = 2$ and $\theta = e^{-1}/2$ for $n = 25, 50$ and $100$ (**e**) Sub-Exponential Self-Repelling Walk (SESRW) with $\beta = 1$ and $\kappa = 0.5$ such that $d_w = 5/3$ and $\theta \approx 0.3$ for $n = 25, 50$ and $100$ (**f**) True Self-Avoiding Walk (TSAW) with $\beta = 1$ such that $d_w = 3/2$ and $\theta = 1/3$ for $n = 25, 50$ and $100$ (**g**) subdiffusive Average Lévy Lorentz (subALL) with $a = 0.25$ such that $d_w = 2.75$, $d_w^0 = 2$ and $\theta = 7/11$ for $n = 10, 100$ and $1000$ (**h**) superdiffusive ALL (supALL) with $a = 0.5$ such that $d_w = 3/2$, $d_w^0 = 2$ and $\theta = 1/3$ for $n = 10, 100$ and $1000$ (**i**) exact rescaled tail distribution (see SI) for scaled Brownian motion (sBm) with $\beta = 0.75$ such that $d_w = 8/3$, $d_w^0 = 2$ and $\theta = 3/8$ for $n = 100, 1000$ and $10,000$. Increasing values of $n$ are represented respectively by blue circles, orange stars and green squares. The black dashed line represents the algebraic decay $\tau^{-\theta}$ while the red dashed line stands for the algebraic decay $\tau^{-1/d_w^0}$.

distance to the origin, and (i) the scaled Brownian motion (sBm)[58] for which the jumping rate is an algebraic function of time, and which is a paradigmatic model of subdiffusion[59].

Figure 2 reveals excellent quantitative agreement between numerical simulations and our analytical results. The data collapse of the properly rescaled record ages tail distribution and the confirmation of the two successive algebraic decays $\tau^{-1/d_w^0}$ and $\tau^{-\theta}$ show that Eq. (1) unambiguously captures the dependence on both the number of records $n$ and the time $\tau$ (further confirmed by the analytical determination of the full tail distribution in the solvable case of the sBm, see SI). We emphasise that the very different nature of these examples (subdiffusive and superdiffusive, aging and non-aging, covering all classes of non-Markovian RWs) shows the broad applicability of our approach.

## Discussion

We demonstrate the relevance of our results by showing that they apply even when the hidden degrees of freedom responsible for the non-Markovianity of the dynamics are unknown, as is the rule in real observations.

This is illustrated by considering both trajectories involving a variety of tracers in complex fluids (see Fig. 3c–e, which provide experimental realisations[26] of several non-Markovian RW models

discussed above) and real time series in diverse fields displaying memory effects, for which record ages are crucial as they characterise the occurrence of extreme events (see Fig. 3a, b and f–h).

Specifically, we consider the following data: (a) river flows[32] $(1/d_w \approx 0.14)$, (b) volcanic soil temperatures[29,30] $(1/d_w \approx 0.42)$, (c) trajectories of microspheres in gels[26] $(1/d_w \approx 0.43)$ (d) trajectories of vacuoles inside an amoeba[26] $(1/d_w \approx 0.67)$, (e) trajectories of telomeres in a nucleus[26,60] $(1/d_w \approx 0.25)$, (f) pyrimidines/purines DNA RW where a step value is given by the nucleotide type, $+1$ for adenine/thymine, $-1$ for cytosine/guanine[34,35] $(1/d_w \approx 0.67)$, (g) cumulative air temperatures[31] $(1/d_w \approx 0.8)$, (h) cumulative Ethernet traffic[36–38] $(1/d_w \approx 0.8)$. The walk dimension $d_w$ was estimated by applying the Detrending Moving Average (DMA) method[61,62] to these data, which removed the deterministic behaviours (see SI for details on the datasets' analysis). Indeed, the characterisation of extreme events, and thus records, requires the meticulous examination of fluctuations around the trend, as underlined in Refs. 31,63.

We stress that we do not require any knowledge on the microscopic details of the process to obtain the record age statistics provided by Eq. (1). In particular, the processes are not necessarily Gaussian and can exhibit various distributions of the increments $x_t \equiv X_{T+t} - X_T$ (see Fig. 3), as long as they are asymptotically scale-invariant (the sampling time of the data is much longer than the

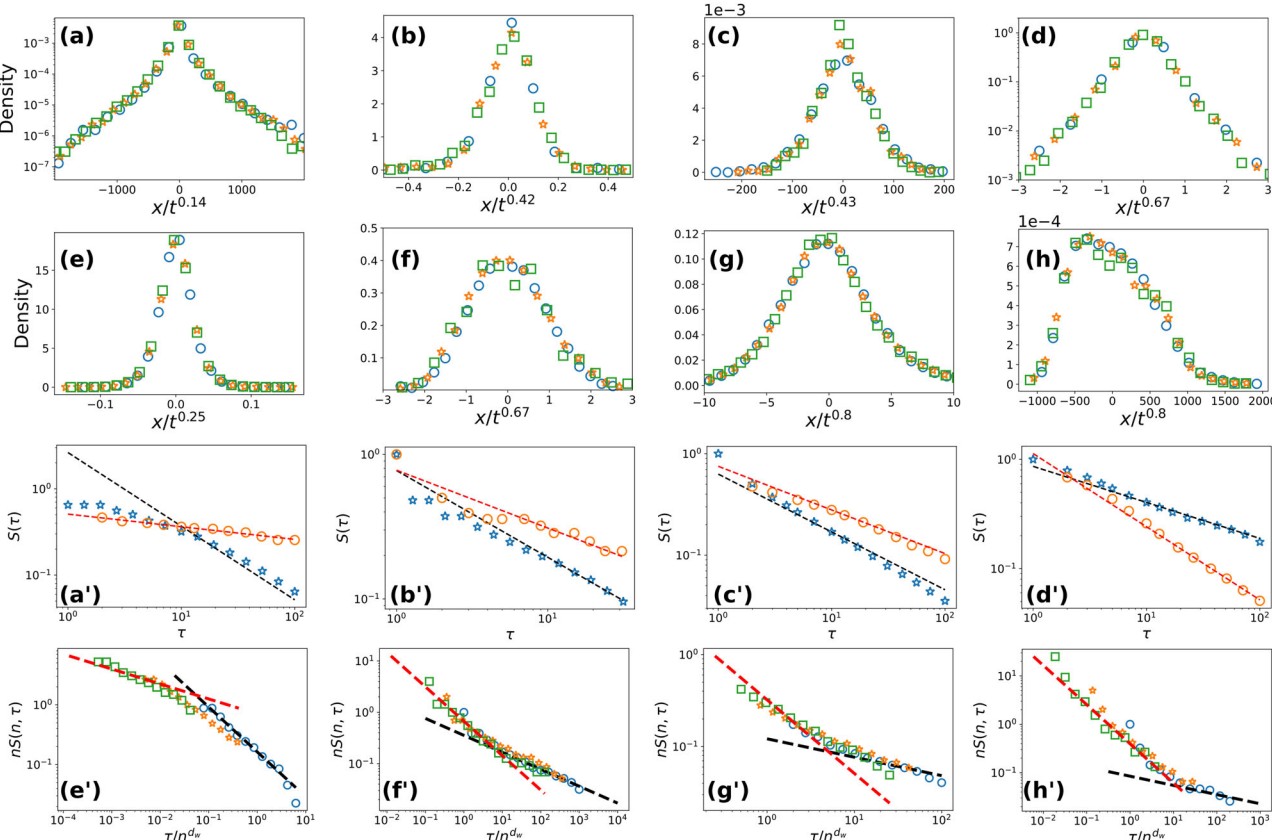

**Fig. 3 | Universal record age distributions for non-Markovian RWs: theoretical predictions (lines) vs experimental RW realisations and real time observations (symbols). a–h** Distribution of the increment $x_t = X_{t+T} - X_T$ at different times $t$ normalised by $t^{1/d_w}$ for: (**a**) river discharge ($t = 10, 20,$ and $40$), (**b**) volcanic soil temperature ($t = 5, 10,$ and $20$), (**c**) motion of microspheres in a gel ($t = 2, 4,$ and $8$), (**d**) motion of vacuoles inside an amoeba ($t = 10, 20,$ and $40$), (**e**) motion of telomeres ($t = 20, 40,$ and $80$), (**f**) DNA RW ($t = 20, 40,$ and $80$), (**g**) cumulative air temperature ($t = 5, 10,$ and $20$), and (**h**) Ethernet cumulative requests ($t = 500, 1000,$ and $2000$). Increasing values of times are represented successively by blue circles, orange stars and green squares. **a′–d′** Statistics of the time to first reach the initial

value in the sub interval (blue stars) and the statistics of the records (regardless of the number $n$ of records, orange circles) for (**a′**) river discharge, (**b′**) volcanic soil temperature, (**c′**) motion of microspheres in a gel, and (**d′**) motion of vacuoles inside an amoeba. The black dashed line represents the algebraic decay $\tau^{-1+1/d_w}$ while the red dashed line stands for the algebraic decay $\tau^{-1/d_w}$. (**e′–h′**) Rescaled tail distribution of record ages $\tau_n$ for different values of the number of records $n$ for (**e′**) motion of telomeres ($n = 1, 3,$ and $6$), (**f′**) DNA RW ($n = 1, 2,$ and $4$), (**g′**) cumulative air temperatures ($n = 1, 2,$ and $3$), and (**h′**) Ethernet cumulative requests ($n = 1, 5,$ and $25$). Increasing values of $n$ are represented by blue circles, orange stars, and green squares. The lines represent the algebraic decays as for (**a′–d′**).

microscopic time scales involved in the process to avoid effects similar to those observed in Ref. [64], as it is checked in Sec. S3 of SI.

Figure [3] demonstrates the quantitative agreement between various real data (see SI Supplementary Fig. 8 for additional datasets, including examples displaying aging of the increments $x_t$) and our analytical predictions given by Eq. ([1]). The strong dependence of record ages on the number $n$ of records already achieved, predicted by our analytical approach and confirmed by both numerical simulations and real observations, is a direct manifestation of the non-Markovian feature of the underlying RWs. These results quantitatively demonstrate the significance of memory effects in the record ages of non-Markovian RWs, providing the tools to better predict record-breaking events.

## Methods
### Numerical simulations of non-Markovian RWs
In this section, we present briefly the models and the numerical methods used to generate the data in Fig. [2].

(**a**) *Fractional Brownian motion (fBm)*. The fBm is a non-Markovian Gaussian process, with stationary increments. Thus, an fBm $X_t$ of Hurst index $H$ is defined by its covariance

$$\mathrm{Cov}(X_t, X_{t'}) = \frac{1}{2}\left( t^{2H} + t'^{2H} - |t - t'|^{2H} \right). \quad (6)$$

The steps $\eta_t = X_t - X_{t-1}$ are called fractional Gaussian noise (fGn). Nowadays, the fBm is broadly spread and its implementations could be found in standard packages of python or Wolfram Mathematica.

(**b**) *Quenched fBm (qfBm)*. This process is an extension of fBm to quenched initial conditions, which results in non-stationary increment statistics. In particular, it describes the height fluctuations under Gaussian noise of an initially flat interface. Then $X_t$ corresponds to the height of the interface at position $x = 0$, $X_t = h(0, t)$, $h(x, t)$ following the Stochastic Differential Equation (SDE)

$$\partial_t h(x, t) = -(-\Delta)^{z/2} h(x, t) + \eta(x, t). \quad (7)$$

Here $\eta(x, t)$ is a Gaussian noise with possible spatial correlations. We solve numerically this SDE with a spatial discretization $\Delta x = 1$ and a time discretization $\Delta t = 0.1$. The system is initially flat, $h(x, t = 0) = 0$.

(**c**) *Elephant RW (eRW)*. This process is representative of interactions with its own trajectory. At time $t$, the step $\eta_t$ is drawn uniformly among all the previous steps $\eta_i$ ($i < t$) and is reversed with probability $\beta$.

(**d**) *Self-attractive walk (SATW)*. This model is a prototypical example of self-interacting RWs. In the SATW model[50–53], the RW at position $i$ jumps to a neighbouring site $j = i \pm 1$ with probability depending on the number of times $n_j$ it has visited site $j$,

$$p(i \to j) = \frac{\exp\left[-\beta H(n_j)\right]}{\exp\left[-\beta H(n_{i-1})\right] + \exp\left[-\beta H(n_{i+1})\right]}, \quad (8)$$

where $H(0) = 0$, $H(n > 0) = 1$ and $\beta > 0$.

(**e-f**) *Exponential self-repelling RW*. This is another example of self-interacting RW. In this model, the RW at position $i$ jumps to a neighbouring site $j = i \pm 1$ with probability depending on the number of times $n_j$ it has visited site $j$,

$$p(i \to j) = \frac{\exp\left[-\beta n_j^\kappa\right]}{\exp\left[-\beta n_{i-1}^\kappa\right] + \exp\left[-\beta n_{i+1}^\kappa\right]} \quad (9)$$

where $\kappa$ and $\beta$ are two positive real numbers.

(**g–h**) *Average Lévy Lorentz gas (ALL)*. We consider a RW on a $1d$ lattice with position dependent reflection or transmission probabilities $r(x)$ or $t(x)$. In the subdiffusive model (resp. superdiffusive model), the transmission coefficient $t(x)$ (resp. reflection coefficient $r(x)$) is taken to be proportional to $|x|^{a-1}$ at large distance $|x|$ from the origin.

### Data analysis
In this section we provide the method developed to determine the walk dimension of the time series presented in Fig. [3] as well as numerical checks of their stationarity.

(i) Walk dimension determination: In order to obtain the walk dimension $d_\mathrm{w}$ in a time series, we apply the Detrending Moving Average (DMA) method[61,62], which consists in evaluating the typical fluctuations in a window of size $\ell$ regardless of any bias or deterministic trend. More precisely, for a dataset $(X_t)_{t=0,\dots,N}$, we consider the windows of size up to $\ell_\mathrm{max}$, compute the window averages $x_t^\ell = \frac{1}{\ell} \sum_{i=0}^{\ell-1} X_{t-i}$, and the typical fluctuation for a window of size $\ell$, $F(\ell) = \sqrt{\frac{1}{N - \ell_\mathrm{max}} \sum_{t=\ell_\mathrm{max}}^{N} (X_t - x_t^\ell)^2}$. When several trajectories are available, we consider the average fluctuation over all the trajectories (for telomeres, vacuoles and microspheres in agarose data). If the data behave as a RW of walk dimension $d_\mathrm{w}$, then $F(\ell) \propto \ell^{1/d_\mathrm{w}}$. We obtain the value of $1/d_\mathrm{w}$ via the DMA method to each dataset.

(ii) Check of stationarity: In order to check that the data are stationary, we compare the MSD obtained from the increments $\{x_t = X_{t+T} - X_T\}_{T \leq N/4, t}$ in the first quarter of the data and the increments $\{x_t = X_{t+T} - X_T\}_{3N/4 \leq T, t}$ in the last quarter of the data.

(iii) Record ages in datasets: Record ages are obtained by starting the subtrajectories at values of $t$ equally spaced at intervals at least 200 time steps long, and observing successive records occurring in the subtrajectory. First return times are obtained by starting the subtrajectories at any value of time.

## Data availability
The simulation data of this study are generated based on the code deposited in a GitHub repository[65] located at https://github.com/LeoReg/RecordAges.

The data of the Hadley Centre Central England Temperature (HadCET) project are available at https://www.metoffice.gov.uk/hadobs/hadcet/. The data of the European Climate Assessment & Dataset (ECA&D) project are available at https://www.ecad.eu/. The volcanic soil temperature data are available at Ref. [30]. River discharge data are available at https://portal.grdc.bafg.de/applications/. The GenBank database is available at https://www.ncbi.nlm.nih.gov/genbank/. The data of traffic traces are available at http://ita.ee.lbl.gov/html/contrib/BC.html. Experimental trajectories of fBm realisations are available upon request by the authors of Ref. [26]. Experimental cell migration trajectories are available upon request by the authors of Ref. [54].

## Code availability
The codes used to generate the simulation data presented in this study as well as the code to analyse the experimental data have been deposited in a GitHub repository located at https://github.com/LeoReg/RecordAges.

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

## Acknowledgements

We thank T. Guérin, N. Levernier, and G. Oshanin for helpful discussions, G. Page for careful reading of the paper, and S. Majumdar for mentioning the similarity between the record age statistics and the statistics of the times between visits of new sites[66,67]. We are thankful to D. Krapf, M. Weiss, F. Taheri and C. Selhuber-Unkel for providing us the experimental trajectories of fBm realisations used in Ref. 26. We thank J. d'Alessandro for providing us the experimental cell migration trajectories analysed in Ref. 54. We acknowledge the data providers in the Hadley Centre Central England Temperature (HadCET) and European Climate Assessment & Dataset (ECA&D) projects. We thank the authors of Ref. 30 for giving access to the volcanic soil temperature data. We acknowledge the Global Runoff Data Centre (GRDC), 56068 Koblenz, Germany for providing the Elbe and Rhône rivers' water debit data. We acknowledge the data providers of the GenBank database, hosted by the National Library of Medicine, as well as Jaenicke T., Diederich K.W., Haas W., Schleich J., Lichter P., Pfordt M., Bach A. and Vosberg H.P. who deposited the specific HUMBMYH7 sequence used in this study. We thank the authors of Ref. 36 for the data of traffic traces.

## Author contributions

O.B., L.R. and M.D. contributed to analytical calculations. L.R. and M.D. performed numerical simulations. All the authors wrote the paper. O.B. conceived the research.

## Competing interests

The authors declare no competing interests.
