## [Peer Review File · Nature Communications]

REVIEWER COMMENTS

Reviewer #1 (Remarks to the Author):

The authors investigate the statistics of the record ages, i.e., the times between two subsequent record-breaking events, for non-Markovian random walks (RW). They assume that the stochastic process becomes for late times scale invariant. They also consider aging in the increments of the process. The main result, given in Eq. 1, gives the asymptotic behaviors of the probability $S(n, \tau)$ that the n -th record age (i.e., the time between the n -th and the $(n+1)$ -th record) is larger than τ . The probability $S(n, \tau)$ has different scaling exponents for intermediate and late times. The authors verify their findings for several RW models as well as data from several sources.

The results presented are interesting and noteworthy. Extremal/record statistics of non-Markov processes is an intrinsically hard problem with many real-world applications. However, as several assumptions are required to derive Eq. 1, the authors should clarify their approach's limits. Reading the title, the abstract, and most of the main text, I had the impression that the results of the paper apply generally to any non-Markov RW. Given the reliance on several strong assumptions, the authors should thoroughly justify these assumptions and offer examples of scenarios where they may not hold.

Comments and questions:

- 1) I suggest that the authors clarify in the title and the abstract that they consider scale-invariant processes.
- 2) Are the authors assuming that the stochastic process is continuous? What happens in the presence of discontinuous jumps, e.g., in the case of a Lévy flight?
- 3) The authors remark that a consequence of their result is that the record age distribution depends explicitly on the number of records already achieved. Isn't this already valid in a minimal model with iid variables?
- 4) The assumption in Eq. (3), i.e., that the resetting ages are independent, is not well motivated. This is very unexpected since the process is non-Markov. The authors only mention that they checked this numerically. The authors should clarify when this assumption breaks down.
- 5) The persistence exponent in Figs. 3 ($a'-d'$) could, in principle, depend on the initial value. How is this initial value chosen for these figures?

Reviewer #2 (Remarks to the Author):

\subsection*{Referee report: Record ages of non-Markovian random walks by L'\eo R'\egnier, Maxim Dolgushev and Olivier B'\enichou}

The authors of the manuscript study the problem of record ages in non-Markovian random walk (RW) processes. Through scaling analysis, the authors identify a universal distribution for the record times, distinct from those found in Markovian dynamics. This distribution is characterized by two different time regimes, each exhibiting unique algebraic decay patterns. To validate their theoretical predictions, the researchers conduct numerical simulations on various non-Markovian random walk models. Notably, they demonstrate the practicality of their findings by applying them to real-world time series data with memory effects. Overall, the combination of theoretical insights and empirical evidence establishes the applicability of their findings to several areas of physics, chemistry and biology; I believe it deserves publication once the comments below have been addressed.

\begin{enumerate}

\item The assumption given in In Eq. (2) has been rigorously proven in general only for the case of IID random variables. This raises the question of whether the correlation between the record times of the studied processes is weak. The authors should address this point in the manuscript and explain why Eq. (2) holds true.

\item In the context of correlated random variables measurement, the authors should consider citing a recent paper by Zarfaty et al. Physical Review L. 129.9 (2022): 094101.

where it is shown that extreme value statistics of simple correlated processes is sensitive to finite discretization of sampling. It is not clear if the results of present paper have similar tendency.

\item Minor points:

\begin{enumerate}

\item On page 4 of the supplementary materials, there is a citation with a question mark.

\item On page 8 of the supplementary materials, the word ``therwords" is missing a space.

\item On page 10 of the supplementary materials, the word ``seasonal" has an extra ``n".

\end{enumerate}

\end{enumerate}

Reviewer #3 (Remarks to the Author):

The authors study the distribution of record ages for general non-Markovian random walks – with a restriction on generality being that the random walk should have symmetrically-distributed jumps, and should converge to a scale-invariant process with almost surely continuous sample paths as well as scale-invariant increments (but potentially aging ones).

Under these assumptions, the authors establish two regimes for the tail of the distribution of the age of the n -th record. These two regimes are universal in the sense that they exist and take the same form independently on more specific details of the jump distribution. These two regimes are characterized by a dependence on n and by exponents that can be explicitly derived from: (i) the walk's limiting scale-invariant behaviour, (ii) the increments' aging exponent, and (iii) the usual persistence exponent.

To appreciate the significance of the result, recall that in the Markovian case, there is only one regime, no dependence on n and the only exponent involved is the usual persistence exponent which in that case is equal to $1/2$.

I think this is a remarkable paper, with far-reaching implications, and yet surprisingly simple to read and follow through (the authors have obviously done a great work at making everything so accessible and clear). I do recommend publication, as is.

We thank all the Referees for their careful reading of the manuscript and constructive comments. We reproduce their reports below in red and our responses are interleaved.

REFEREE 1

The authors investigate the statistics of the record ages, i.e., the times between two subsequent record-braking events, for non-Markovian random walks (RW). They assume that the stochastic process becomes for late times scale invariant. They also consider aging in the increments of the process. The main result, given in Eq. 1, gives the asymptotic behaviors of the probability $S(n, \tau)$ that the n -th record age (i.e., the time between the n -th and the $(n + 1)$ -th record) is larger than τ . The probability $S(n, \tau)$ has different scaling exponents for intermediate and late times. The authors verify their findings for several RW models as well as data from several sources. The results presented are interesting and noteworthy. Extremal/record statistics of non-Markov processes is an intrinsically hard problem with many real-world applications. However, as several assumptions are required to derive Eq. 1, the authors should clarify their approach's limits. Reading the title, the abstract, and most of the main text, I had the impression that the results of the paper apply generally to any non-Markov RW. Given the reliance on several strong assumptions, the authors should thoroughly justify these assumptions and offer examples of scenarios where they may not hold.

We have followed all the referee's comments and changed our manuscript accordingly. In the revised version we state more explicitly that our results rely on three hypotheses: asymptotic scale-invariance, continuity, and non-smoothness of the process. Under these hypotheses, we now provide a new analytic justification that the record ages τ_k can be treated as effectively independent (see the new section S1.D of the SI).

As soon as at least one of the hypotheses breaks, we expect our results not to hold anymore. To illustrate this, we provide now classical examples of stochastic processes, which are not covered by the above-mentioned hypotheses, lying hence beyond the scope of our work (Lévy flights and random acceleration processes).

Comments and questions:

1) I suggest that the authors clarify in the title and the abstract that they consider scale-invariant processes.

We followed the suggestion of the Referee and changed the title and the abstract of the paper.

2) Are the authors assuming that the stochastic process is continuous? What happens in the presence of discontinuous jumps, e.g., in the case of a Lévy flight?

As noted by the Referee, we consider continuous processes in our work. To stress this, the continuity hypothesis is now mentioned in the abstract.

This assumption is crucial in order to map the time to break n records to the time to reach level $n\Delta x$ (see section S1.C of SI). In the presence of discontinuous jumps, this mapping breaks down and our results are not expected to hold anymore: For example, in the case of 1d Lévy flights with i.i.d. jumps, distributed according to $\mathbb{P}(\eta_t = \eta) \propto |\eta|^{-1-\alpha}$, the time τ_k to break the k^{th} record is independent of k and given by the Sparre-Andersen result, $S(k, T) \sim 1/\sqrt{\pi T}$ (see Ref. [1]); in this example $d_w = \alpha$, while the record-age exponent of the survival probability is independent of α . We stress now this point in the revised version (see footnote [54]).

3) The authors remark that a consequence of their result is that the record age distribution depends explicitly on the number of records already achieved. Isn't this already valid in a minimal model with i.i.d. variables?

We agree that for i.i.d. variables $(X_t)_{t \geq 0}$ the record-age distribution also depends on the number of records previously achieved (as can be found in [2]). However, in this case, the variables $(X_t)_{t \geq 0}$ do not satisfy the evolution equation $X_{t+1} = X_t + \eta_{t+1}$ considered in our work. This point is now explicitly commented in the main text (see footnote [55]).

4) The assumption in Eq. (3), i.e., that the resetting ages are independent, is not well motivated. This is very unexpected since the process is non-Markov. The authors only mention that they checked this numerically. The

authors should clarify when this assumption breaks down.

In addition to our extensive numerical check, we now propose an analytical justification of the effective independence of record ages based on the criteria given in [3], see the new section S1.D of SI. It shows that the scale-invariance, the continuity, and the non-smoothness are the only hypotheses needed to neglect the correlations.

Specifically, the criteria consists in the comparison of the typical fluctuations of the maximum of the τ_k with the correlations between two adequately distant record ages. Relying on the three above-mentioned hypotheses on the process, we show that the correlations between the record ages are small compared to the fluctuations of the maximum record age. In other words, this shows that record ages can be considered as effectively independent in Eq. (3).

This argument also highlights why the hypotheses of scale-invariance, continuity, and non-smoothness are required in the derivation:

- asymptotic scale invariance is needed in our estimates of the typical fluctuations of the time to break the n^{th} record as well as the scaling form for the record age distribution;
- continuity is required for the mapping between the time to break n records to the time to reach level $n\Delta x$;
- non-smoothness is essential for having small fluctuations of the record ages as compared to the ones of the maximum.

In the revised version we followed the recommendation of the Referee and explicitly mention processes, where our results do not apply due to breaking of one of the above-mentioned hypotheses:

- Lévy flights are discontinuous;
- The Random Acceleration Process is smooth.

5)The persistence exponent in Figs. 3 (a'-d') could, in principle, depend on the initial value. How is this initial value chosen for these figures?

The increments of data used in Figs. 3 (a'-d') are stationary (which is checked in Supplementary Figure 6 on the third and fourth lines), what fixes the persistence exponent. The initial value is taken at every time point of the data. We now indicate this in the SI.

REFEREE 2

The authors of the manuscript study the problem of record ages in non-Markovian random walk (RW) processes. Through scaling analysis, the authors identify a universal distribution for the record times, distinct from those found in Markovian dynamics. This distribution is characterized by two different time regimes, each exhibiting unique algebraic decay patterns. To validate their theoretical predictions, the researchers conduct numerical simulations on various non-Markovian random walk models. Notably, they demonstrate the practicality of their findings by applying them to real-world time series data with memory effects. Overall, the combination of theoretical insights and empirical evidence establishes the applicability of their findings to several areas of physics, chemistry and biology; I believe it deserves publication once the comments below have been addressed.

We have followed all the referee's comments and changed our manuscript accordingly.

The assumption given in In Eq. (2) has been rigorously proven in general only for the case of IID random variables. This raises the question of whether the correlation between the record times of the studied processes is weak. The authors should address this point in the manuscript and explain why Eq. (2) holds true.

We propose now a justification of the effective independence of record ages based on the criteria given in [3], see the new section S1.D of SI. It shows that the scale-invariance, the continuity, and the non-smoothness are the only hypotheses needed to neglect the correlations, see also the answer to comment 4) of the 1th referee for details, which goes in line with this comment.

Based on this result, as the Referee points out, we come to Eq. (2), as it holds for independent random variables.

In the context of correlated random variables measurement, the authors should consider citing a recent paper by Zarfaty et al. *Physical Review L.* 129.9 (2022): 094101. where it is shown that extreme value statistics of simple correlated processes is sensitive to finite discretization of sampling. It is not clear if the results of present paper have similar tendency.

We thank the referee for pointing out this work. Indeed, in our work we use the extreme value statistics (EVS) of the record ages, $\max(\tau_0, \dots, \tau_{n-1})$, and one might wonder how the sampling might affect this statistics. However, we consider only scale invariant processes in this work, meaning that there is no additional length scale that can be associated with the time discretization, as is the case for the Ornstein-Uhlenbeck process considered in the work of Zarfati et al. Thus, our results on the random variables τ_k will remain the same for sampling at different time steps, as the processes do not change when changing the time scale (which is very different from, e.g., the Ornstein-Uhlenbeck process).

Besides, one has to be careful with the real data for which an additional time scale related to a break of the scale invariance might exist that can lead in turn to the effects described in the work of Zarfaty et al. However, these effects are absent for the data considered in our work, given their scale invariance, see the first two lines of Fig. 3 and the third and fourth lines of Supplementary Figure 6. A comment on this point is now present at page four of the main text.

To further illustrate this point, one can consider as an example the case of a $1d$ run-and-tumble particle where steps η_t have a finite correlation time (in a similar manner as the Ornstein-Uhlenbeck process was considered to study random variables X_t with finite correlation time). In this example, the statistics of the τ_k is exactly given by the survival probability of the run-and-tumble particle with correlation time T not to cross Δx starting from the origin with positive initial speed, due to Markovianity of the couple position and speed. As shown in [4], the survival probability at large times is the same as the one of its scaling limit, the $1d$ Brownian motion. This emphasizes that when looking at records statistics in the scale invariant regime, the microscopic times are not relevant anymore.

Minor points:

- On page 4 of the supplementary materials, there is a citation with a question mark.
- On page 8 of the supplementary materials, the word “therwords” is missing a space.
- On page 10 of the supplementary materials, the word “seasonal” has an extra “n”.

We corrected the points raised by the referee.

REFEREE 3

The authors study the distribution of record ages for general non-Markovian random walks – with a restriction on generality being that the random walk should have symmetrically-distributed jumps, and should converge to a scale-invariant process with almost surely continuous sample paths as well as scale-invariant increments (but potentially aging ones).

Under these assumptions, the authors establish two regimes for the tail of the distribution of the age of the n -th record. These two regimes are universal in the sense that they exist and take the same form independently on more specific details of the jump distribution. These two regimes are characterized by a dependence on n and by exponents that can be explicitly derived from: (i) the walk’s limiting scale-invariant behaviour, (ii) the increments’ aging exponent, and (iii) the usual persistence exponent. To appreciate the significance of the result, recall that in the Markovian case, there is only one regime, no dependence on n and the only exponent involved is the usual persistence exponent which in that case is equal to $1/2$.

I think this is a remarkable paper, with far-reaching implications, and yet surprisingly simple to read and follow through (the authors have obviously done a great work at making everything so accessible and clear). I do recommend publication, as is.

We are pleased to know that the referee recommends to publish our manuscript as is and we are very thankful.

[1] S. N. Majumdar and R. M. Ziff, Universal record statistics of random walks and Lévy flights, *Phys. Rev. Lett.* **101**, 050601 (2008).

- [2] C. Godrèche, S. N. Majumdar, and G. Schehr, Record statistics of a strongly correlated time series: random walks and Lévy flights, *J. Phys. A: Math. Theor.* **50**, 333001 (2017).
- [3] D. Carpentier and P. Le Doussal, Glass transition of a particle in a random potential, front selection in nonlinear renormalization group, and entropic phenomena in Liouville and sinh-Gordon models, *Phys. Rev. E* **63**, 026110 (2001).
- [4] P. Le Doussal, S. N. Majumdar, and G. Schehr, Noncrossing run-and-tumble particles on a line, *Phys. Rev. E* **100**, 012113 (2019).

REVIEWERS' COMMENTS

Reviewer #1 (Remarks to the Author):

I thank the authors for the revised version of the manuscript. My comments have been thoroughly taken into account. I recommend the publication of the manuscript in its current form.

Reviewer #2 (Remarks to the Author):

The authors have addressed the concerns and suggestions raised in my initial review, resulting in a significantly improved manuscript. In light of the revisions by the authors, I am inclined to recommend the acceptance of the paper for publication in Nature Communications. The revised manuscript presents a valuable contribution to the scientific community and will undoubtedly stimulate further discussions and advancements in the field.

We thank all the Referees for their careful reading of the manuscript and constructive comments. We reproduce their final comments below in red.

REFEREE 1

I thank the authors for the revised version of the manuscript. My comments have been thoroughly taken into account. I recommend the publication of the manuscript in its current form.

REFEREE 2

The authors have addressed the concerns and suggestions raised in my initial review, resulting in a significantly improved manuscript. In light of the revisions by the authors, I am inclined to recommend the acceptance of the paper for publication in Nature Communications. The revised manuscript presents a valuable contribution to the scientific community and will undoubtedly stimulate further discussions and advancements in the field.

We thank both Referees for their positive assessment of our work and their recommendation in the publication of our results.